# High-Intensity Functional Training: Perceived Functional and Psychosocial Health-Related Outcomes from Current Participants with Mobility-Related Disabilities

**DOI:** 10.3390/sports11060116

**Published:** 2023-06-12

**Authors:** Lyndsie M. Koon, Jean P. Hall, Kristen A. Arnold, Joseph E. Donnelly, Katie M. Heinrich

**Affiliations:** 1Research and Training Center on Independent Living, Life Span Institute, University of Kansas, 1000 Sunnyside Ave., Room 1052, Lawrence, KS 66045, USA; jhall@ku.edu; 2Institute for Health and Disability Policy Studies (KU-IHDPS), Life Span Institute, University of Kansas, 1000 Sunnyside Ave., Room 1052, Lawrence, KS 66045, USA; 3Department of Special Education, School of Education, University of Kansas, 1122 W. Campus Rd., Lawrence, KS 66045, USA; kristen.arnold@ku.edu; 4Division of Physical Activity and Weight Management (DPAWM), Department of Internal Medicine, University of Kansas Medical Center, 3901 Rainbow Boulevard Mailstop 1007, Kansas City, KS 66160, USA; 5Department of Kinesiology, College of Health and Human Sciences, Kansas State University, Manhattan, KS 66506, USA; 6Department of Research and Evaluation, The Phoenix, Denver, CO 64035, USA

**Keywords:** mobility disabilities, community-based exercise, inclusive, health, function, psychosocial, high-intensity functional training

## Abstract

Background: People with mobility-related disabilities (MRDs) experience many personal and environmental barriers to engagement in community-based exercise programs. We explored the experiences of adults with MRD who currently participate in high-intensity functional training (HIFT), an inclusive and accessible community-based exercise program. Methods: Thirty-eight participants completed online surveys with open-ended questions, with ten individuals also participating in semi-structured interviews via telephone with project PI. Surveys and interviews were designed to examine changes to perceived health, and the elements of HIFT that promote sustained participation. Results: Thematic analysis revealed themes related to health changes following HIFT participation including improved physical, functional, and psychosocial health outcomes. Other themes emerged within the HIFT environment that promoted adherence for participants such as accessible spaces and equipment, and inclusive HIFT sessions and competitions. Additional themes included participants’ advice for the disability and healthcare communities. The resulting themes are informed by the World Health Organization’s International Classification of Functioning, Disability, and Health. Conclusion: The findings provide initial data on the potential effects of HIFT on multiple dimensions of health outcomes and contribute to the growing literature on community-based programs that are adaptable and inclusive for people with MRD.

## 1. Introduction

The benefits of exercise for individuals with mobility-related disability (MRD) are well documented [1,2,3,4]. However, the majority of exercise trials reviewed (84%) have been conducted in clinical, laboratory, or rehabilitation settings, and often utilize equipment or programming that is unavailable at most community-based exercise facilities [1]. When considering community-based exercise programs, individuals with MRD and other disabilities face unique barriers to participation, including but not limited to inaccessible facilities and equipment, lack of support or knowledge from fitness professionals, transportation, financial, or time-based barriers, as well as personal challenges such as pain or fatigue [5,6,7,8,9,10,11,12]. Additionally, healthcare professionals indicate a lack of knowledge relative to safe, effective, and inclusive community-based exercise programs for their clients [13,14]; thus, pharmacological or surgical approaches are prescribed far more frequently than exercise [15]. Outpatient physical and occupational therapy provides temporary exercise opportunities; however, these services are typically prescribed for short-term rehabilitation with limitations on the total number of visits and are not a long-term strategy for activity engagement [16]. Sustainable community-based programs that facilitate activity engagement beyond the research or rehabilitation setting are greatly needed for this population [17,18]. The evaluation of aneffective community-based exercise programs would allow for recommendations for exercise in both the disability and healthcare communities. This paper reports findings from a study of perceived health changes among people with MRD who participate in a community-based exercise program and key elements of the environment that promote participation. 

### 1.1. High-Intensity Functional Training (HIFT)

High-intensity functional training (HIFT) is one of the fastest growing fitness trends worldwide [19,20], with over 15,000 licensed facilities in 150 countries [21,22] and many other non-licensed facilities. HIFT incorporates multiple modalities of varied functional movements designed to support performance in everyday activities, such as picking up a heavy object from the ground, sit-to-stand movements, and reaching overhead. Exercises that address these functional movements are especially relevant for individuals with MRD who use them to transfer, pick up heavy objects from a seated position, or ambulate without supports. 

Although the sessions come in a prescribed manner, every aspect is modifiable, and individuals can choose among a multitude of options that scale the workout to fit their needs or capabilities. This unique feature allows for participants to self-regulate the session intensity, ensuring any individual, regardless of age, gender, fitness level, or impairment, can participate. Among non-disabled HIFT participants, research results indicate greater enjoyment (arousal and pleasure) [23] and feelings of autonomy, self-empowerment, and control [24,25,26] due to the self-regulation opportunities. Additionally, the options for modification present a consistent opportunity for skill development, requiring the participant to focus on incremental improvements of their individual performance—a distinctive aspect that has been found to promote perceptions of competence and autonomy among non-disabled HIFT participants [27,28] and foster long-term engagement [24,26,29,30,31,32]. 

HIFT was developed as an inclusive program with an objective to support community building [31]. Participants are encouraged to socially engage with others during warm-up and cool-down periods; include new members; and regardless of scaling, the workout is completed concurrently in a group-setting. This social environment has been found to promote the sense of social support, affiliation, and perceptions of relatedness [26,29,33,34], and greater perceived value in the activity [35]. These psychosocial constructs fostered by the exercise environment have been found to promote positive behaviors such as persistent engagement and long-term adherence [23,36]. For people with MRD, these environmental psychosocial factors may be especially important, as exercise interventions that support self-regulation, learning, and social engagement lead to sustained engagement [4,37], and many of these health benefits only occur with long-term adherence [17,18]. 

Research among non-disabled populations indicates significant improvements to various components of physical and functional health, including body composition, strength, endurance, anaerobic and aerobic capacity, balance, power, and flexibility, following HIFT interventions compared to other types of exercise (e.g., HIIT) [23,38,39,40,41,42,43,44,45]. Data on the effects of HIFT among adults with MRD are limited to pilot studies and unpublished theses or dissertations with poor descriptions of the intervention and un-powered sample sizes [46,47,48,49]. Collectively, however, the studies show feasibility, including high retention rates, positive intervention experiences, improvements to components of functional capacity (e.g., walking speed and balance) [48,49], and improved confidence in performing various functional movements (e.g., sit to stand) [46,47]. 

Anecdotal evidence suggests that people with disability have been adapting and participating in HIFT for more than a decade. For example, the Adaptive Training Academy (ATA) was established in 2012 by adaptive fitness leaders to teach movements adaptations, inclusive class strategies, and safety and risk mitigation to anyone looking to adapt HIFT for people with disability. The goal of the program is to reduce the limiting effects of the impairment while increasing overall function, and to advance accessibility and inclusion of HIFT through education [50]. To date, ATA has provided the Adaptive and Inclusive Training course to more than 3600 coaches and 1575 gyms and organizations representing 51+ countries. Other anecdotal evidence shows that various national and international HIFT competitions have experienced a 120–133% increase in competitors with disabilities from 2016–2020. Meanwhile, a popular international HIFT competition hosted adaptive divisions in 2020 and 2021, with more than 1000 competitors with disabilities registering, and the National Veterans Wheelchair Games introduced a HIFT division starting in 2021. These opportunities to compete have been cited as a motive for exercise adherence among people with disability [51,52]. Empirical evidence exploring the experience of people with disabilities in HIFT is missing and, thus, informed the approach to the current study.

### 1.2. Study Purpose

Clinical research supports claims that exercise programs can have a significant impact on health, function, and quality of life for people with disabilities [1,4,53]. However, not represented in this literature is the evaluation of existing community-based exercise programs that reduce disparities in access to, and inclusion in, sustained exercise for people with MRD outside of the clinical setting. Given its prevalence across the country, HIFT has high potential for dissemination and reach. The disabled community is already engaging in HIFT yet, little evidence exists to show the effects of this program on health outcomes. Thus, the purpose of this study is to evaluate the perceived health outcomes from HIFT participation, as well as motives for initial and continued engagement among *current* HIFT participants with MRD. 

## 2. Methods

We employed a participant-oriented, phenomenological approach [54] using surveys and semi-structured interviews. Additionally, we utilized the conceptual framework of the World Health Organization’s International Classification of Functioning, Disability, and Health Model (ICF) [55] with these exploratory methods to inform the results, allowing for a more organized and deeper understanding of the health outcomes and environmental factors that influence participation and adherence to HIFT [56].

### 2.1. Study Design and Approach

Study methods include a qualitative description using data collected from open-ended survey responses and semi-structured interviews to understand the subjective experiences of current HIFT participants with MRD. Given the paucity of empirical evidence in this area, a qualitative approach allowed for a deeper exploration of the contextual factors within HIFT that influence participants’ unique perspectives and experiences of a novel phenomenon [57]. Semi-structured interviews provided the opportunity for an in-depth exploration of participants’ experiences with HIFT, and retrospective self-reported effects on their health and function. Further, the qualitative descriptive analysis provides a comprehensive summary of the health outcomes and motives for participation, and reduces the need for researcher interpretation of the data [58]. This retrospective approach is helpful when evaluating subjective health outcomes, as it allows participants to respond based on the knowledge gained from their experience with HIFT, resulting in a more accurate assessment of health changes [59,60]. The Standards for Reporting Qualitative Research (SRQR) were used to provide transparency in all aspects of our methodology [61]. 

### 2.2. Conceptual Framework

Understanding what makes an intervention appropriate for participants requires a thorough examination of its perceived effects on health in terms of body function and structures and exploring the larger impact it may have on functional independence, participation in work, leisure activities, and social engagement. We selected the ICF model for this study because it is a framework describing how various constructs interact and affect health conditions at the personal (e.g., body structure and function) and societal (e.g., relationships and participation) levels (see Figure 1) [55]. The ICF framework has been widely accepted as a method to facilitate the analysis of qualitative data [62]. More specifically, the ICF has been effectively used to identify factors that empower or prevent people with disability from participating in physical activity in the community [63] and to examine the effects of an exercise intervention among participants with physical disability [64].

### 2.3. Participant Recruitment, Criteria, and Sampling

Participants were recruited to the study via HIFT listservs, social media, and newsletters. Those who were interested contacted study staff, screened for eligibility via telephone, and were confirmed eligible if they met the following criteria: (a) 18+ years of age; (b) identified as having a permanent disability that affects their mobility; (c) served as their own guardian and were able to provide informed consent; and (d) had been participating in HIFT for at least 3 months at the time of data collection. Eligible participants (N = 38) completed the initial online survey via Qualtrics. Among those who completed the survey, convenience sampling occurred for participation in individual, semi-structured interviews (*n* = 10). Interview recruitment concluded after ten participants had completed interviews, as previous studies have shown that a sample size of just six may be sufficient to reach saturation for a phenomenological approach [65]. 

### 2.4. Instruments

The online survey contained items related to demographics, disability type, and difficulty with functions or activities [66]; HIFT experience (e.g., length and frequency of participation); and open-ended questions to explore perspectives on the effects HIFT participation has had on physical health and functional independence, community engagement, quality of life, and advice for key stakeholders. 

Semi-structured interview scripts were developed iteratively with three members of the research team to elicit further details regarding experiences and perceptions with HIFT and self-reported changes to physical, functional, and psychosocial health, as well as to explore experiences and interactions within the HIFT environment [67,68]. The questions broadly asked participants to report on the effects of HIFT engagement on their health and function, with follow-up questions regarding changes to disability-related or general health, functional independence, use of assistive devices, or engagement in their community, and to understand their motives for initial and sustained HIFT engagement. Additional questions were developed to explore any information participants would want the disabled or healthcare community to know about their experiences with HIFT, and any recommendations they would have for others with similar disability who are hesitant to try HIFT. Interviews were flexible to participant responses, and prompts were used to gain clarification. Informed consent was obtained from all participants prior to completing the online surveys, and again prior to participation in the interviews. Participants who completed the interviews were compensated USD 40. 

### 2.5. Data Collection and Analysis

Data collection and analysis were conducted concurrently between November 2020 and July 2021. Participants who completed the survey and indicated interest in participating in further research activities were contacted and provided with more information about the interview portion of the study. Individual interviews were conducted by the lead investigator, who had no relationship with the participants prior to data collection but had extensive experience conducting qualitative studies among people with disabilities. Interview sessions lasted approximately 45–60 min, were conducted via telephone or teleconference platform (e.g., Zoom) based on participant preference, and audio was recorded. 

The data included in the analysis were participant demographics and other descriptives, open-ended survey responses, and transcripts from the individual interviews. Audio files from the interviews were transcribed verbatim, and identifying information (e.g., participant’s name) was removed from the final transcript to maintain confidentiality. An iterative thematic analysis was conducted as informed by Braun and Clark [69] to identify, organize, assess, and report patterns within a given data set. This approach allowed the investigator to identify broad themes related to perceived changes in physical, functional, and mental, and psychosocial health outcomes experienced as a result of HIFT participation. Next, subthemes were identified for each of the higher order themes that best classified and represented the data. Themes and subthemes were given working definitions and example quotes from the data and shared with a senior author on the research team. The lead author simultaneously applied themes and subthemes to sample transcripts and survey responses using a qualitative software program (Maxqda, VERBI Software 2020). Discrepancies in the segments were shared with the secondary author for clarification and theme application, definitions for the themes and subthemes were explored, questioned, and debated, and alternative themes and subthemes were discussed until a consensus was reached. This iterative approach facilitated the systematic organization and analysis of the data and reduced the interference of researcher biases [70,71]. 

## 3. Results

The results include descriptions of the participants who responded to surveys and engaged in interviews; qualitative themes and subthemes identified concerning health outcomes experienced by HIFT participants with disabilities; key environmental features of HIFT that support initial and sustained participation; and advice for key stakeholders.

### 3.1. Description of Participants

A total of 28 participants completed the online surveys only. The majority were female (54%), with a mean age of 37 years, and reported participating in HIFT for an average of 4.2 years (SD = 2.9 years; range, 3 months to 12 years). A total of 10 participants completed the online survey and the interviews. This subsample was predominantly female (60%), with a mean age of 45 years, and reported participating in HIFT for an average of 3.3 years (SD = 2.4 years; range, 7 months to 9.1 years). See Table 1 for complete participant descriptives. 

### 3.2. Qualitative Results

A thematic analysis identified major themes and subthemes that represent multiple dimensions of participants’ health, including physical, functional, and psychological health outcomes. These themes are presented in relation to the ICF Functioning and Disability domain that describes body structure and function, activities at the person level, and participation at the person-in-society level. The analysis further identified another major theme related to psychosocial health outcomes because of HIFT participation, with multiple subthemes regarding the perceived social support from other HIFT members and the training staff, the connections made with others with disability through social media platforms, integrated classes, and the accessibility and inclusivity of HIFT competitions. 

The ICF domain of contextual factors or, more specifically, environmental factors that emerged from the analysis were related to the physical HIFT environment (e.g., equipment and spaces). Other themes describe participants’ desire for key stakeholders (e.g., medical providers and the disability community) to know about their HIFT experiences and the health outcomes from their participation. 

### 3.3. Health Themes

This section includes the physical, functional, psychological, and psychosocial health outcomes related to HIFT participation. Themes are presented and discussed in relation to the domains of the ICF model. 

#### 3.3.1. Physical and Functional Health Outcomes

Clear themes were identified in the data that related to the ICF domain of functioning and disability (see Figure 2). While body structure and function are oftentimes inextricably linked, we were careful not to draw inferences, and instead present the themes separately to best represent first the “what” of the body structures that reportedly changed due to HIFT engagement, followed by descriptives of the effects of structural changes on functional performance, or the “how”.

Changes to physical health were described in detail and often related to conditions secondary to the participants’ disability, or other aspects of their physical health not related to the disability. Examples include changes to body composition, pain, strength, energy, stamina or fatigue, blood chemistry or other metabolic testing, posture, or even bowel/bladder function. Participants further elaborated on the delay, reversal, or overall prevention of the onset of chronic conditions related to the primary disability and physical inactivity. These structural changes were reported as anecdotal (e.g., perceived changes in fatigue) and evidential as evaluated by objective assessments (e.g., weight) or healthcare provider results (e.g., imaging; blood draws). 

Changes to functional health emerged as the most frequently reported health outcome from HIFT participation. Many of the reported changes to functional capacity/performance were likely due to physical bodily changes; however, most participants did not make that direct link. Instead, changes to function and decreased difficulty with daily activities were reported. It was evident from the interviews that HIFT participation affected levels of functioning, which had a subsequent effect on participation in various activities such as mobility, self-care, housekeeping, or moving objects independently. Example quotes for structural and functional changes can be found in Table 2. 

#### 3.3.2. Psychological Health Outcomes

Participants reported various changes to their psychological health as a result of HIFT participation. According to the ICF model, contextual factors, comprising environmental and personal factors, also affect a person’s body function and structure in addition to their ability and willingness to participate. While environmental factors encompass those outside the individual, personal factors are represented by an individual’s motivation, behavior, and past or current experiences (see Figure 3). 

In this case, the environmental context and effect of HIFT participation expanded beyond that of physical and functional health, as evident by participant reports of changes to psychological health factors such as depression, anxiety, or stress; improved confidence; coping skills; and overall quality of life. The ICF model does not include an exhaustive list of personal factors within the classification system simply due to the diversity of social, cultural, age-related, or gender differences between individuals. Example quotes for changes to psychological health outcomes can be seen in Table 3. 

#### 3.3.3. Psychosocial Well-Being Themes

An extension of the ICF domain of contextual factors includes supports and relationships, attitudes of others and services, systems and policy, and was reflected in the perceived social support participants reported from HIFT engagement (see Figure 4). The major goals of HIFT are to promote a strong sense of community among members and between trainers and members, as well as to facilitate social engagement outside of classes in the form of fundraisers, competitions, and other social events related, or unrelated, to HIFT [23,72,73]. Data revealed frequent reports of participant perceptions of belonging and social support, thus affecting personal factors such as internal mindsets and interactions with others in their community. These data indicate that a common bond is formed between HIFT members, regardless of disability status or other facets of life, both within and outside of the HIFT context. 

Several subthemes emerged related to changes in participants’ psychosocial health. These include the attitudes and willingness of coaches to take on program adaptations and assist the participant, regardless of any previous experience, training, certifications, or knowledge of the individuals’ disability types or needs. The interviews revealed that for many participants, the path to engaging in HIFT often involved collaboration between themselves and their trainers to learn movement and activity adaptations that retained the intended stimulus of the session [50], often accomplished through trial and error. Participants also discussed the use of social media to connect with and learn from other HIFT participants with disabilities across the globe. This contextual factor promoted camaraderie between individuals with similar impairments, as they shared their lived experiences and similar adaptations for a variety of functional movements while engaging in HIFT. Participants further described the accessibility and inclusivity of HIFT competitions, both adaptive-only competitions, as well as access and inclusion in non-adaptive competitions. While most participants described starting out their HIFT experience one on one with a HIFT coach, many had since transitioned to an integrated class setting, in which they complete the HIFT sessions alongside their non-disabled peers. Within integrated class settings, participants performed the HIFT session with the intent to achieve the same stimulus, regardless of their impairment type. Those integrated class settings were described as pivotal in changing societal perceptions about disability, and many of the participants reported being treated “no differently than any other [participant]” in the class. Several participants had even become HIFT trainers, working with people with and without disabilities at their facility, as well as advising on adaptations to the sessions, and competition programming due to the expanding adaptive HIFT competitions. Table 4 presents example quotes from the rest of these subthemes identified under the higher-order theme of psychosocial health. 

### 3.4. Contextual Factor Themes 

The ICF model includes components of contextual factors, including environmental and personal factors. As mentioned, the ICF model does not include an exhaustive list of personal factors within the classification system simply due to the various individual differences (e.g., social, cultural, age-related). Here, we focus on environmental actors within the HIFT that were discussed, which t promote inclusivity and accessibility. These include aspects of the physical environment such as accessible spaces, entryways, and bathrooms, or accessible equipment. Furthermore, interviews inquired about the advice the participants would give to the medical community and/or others with disabilities, regarding their experience with HIFT. This theme is congruent with the ICF domain of environmental factors, specifically regarding systems, policies, attitudes of others, and relationships.

#### 3.4.1. The HIFT Environment

The HIFT environment is well known for its minimalist approach to fitness [74]. Unlike many community-based fitness facilities that provide extensive exercise equipment, such as cardiorespiratory machines (e.g., treadmills) or weight machines, which are often inaccessible to people with MRD, HIFT facilities typically prioritize empty floor spaces.. The large space allows for more freedom of movement, the inclusion of unconventional equipment that can be used in a multitude of ways (e.g., ropes, rigs, boxes, kettlebells), and are adaptable and, therefore, accessible. Because of their settings and minimal use of machines, HIFT spaces (e.g., entries and bathrooms) and exercise equipment are often accessible and customizable to fit the needs and capabilities of the individual, making HIFT an accessible environment regardless of impairment. Another subtheme that emerged is that HIFT participants with disability have learned to develop or use products to assist with movements, which may be conventional (e.g., adaptive hands) or unconventional (e.g., abdominal pad as a lap mat and belts for handles). In some cases, the unconventional use of products has prompted exercise equipment manufacturers to design and market new devices. These contextual factors that allow for the accessibility of community-based exercise facilities have been found to facilitate long-term participation in exercise for people with disabilities [75]. See Figure 5 for a diagram of the relationship between the ICF contextual factors (environment), and Table 5 for example quotes from participants on the physical environment of HIFT. 

#### 3.4.2. Advice for Others 

Surveys and interviews inquired about participants’ advice to others, particularly key stakeholders, such as medical professionals or others with disability, regarding their experience with HIFT.. Participants expressed a desire to increase public knowledge about the effects of HIFT on their health, particularly secondary conditions related, or unrelated, to their primary disability. Participants reported the common misunderstanding about—or the ignorance of- HIFT from others, particularly those in the medical community. They described the perpetual fear that is associated with doing what some may consider “unconventional” exercise, despite the functional properties associated with HIFT movements. Advocating for HIFT as an option for exercise during or following rehabilitation therapy, especially for those who acquired a disability through injury (e.g., SCI), was a common expression. Table 6 provides example quotes from participants on advice for others based on their own HIFT experience. 

## 4. Discussion and Implications

Sufficient research exists to support positive health outcomes from HIFT engagement for individuals without disability. This study aimed to add to the literature by exploring the effects of HIFT engagement on perceived health outcomes among current HIFT participants with MRD and identifying environmental factors that influence HIFT engagement. The thematic analysis yielded health-related themes perceived to be affected by HIFT participation, including physical, functional, psychological, and psychosocial health outcomes. Additional themes related to the contextual factors of the HIFT environment that motivated participation were identified, as well as themes regarding advice for other stakeholders were discussed. 

Participants identified improvements to various physical and functional health outcomes such as improved body composition or strength. Although subjective, these perceived changes concur with findings from previous HIFT interventions for people without disability [23,38]. Disability-specific changes reported by this sample were also congruent with previous studies assessing objective health outcomes following non-HIFT interventions for people with physical disability [1,3,44], such as improved posture or better bowel and bladder function, and were often accompanied by reports of improved objective health measures, such as blood panel assessments or imaging results from healthcare services. Health improvements from HIFT had consequences on other aspects of participants’ functional capacity, including decreased reliance on others, independent mobility, and increased engagement in everyday activities both inside and outside the home. For example, strength gains made through HIFT resulted in easier transfers and moving heavy objects independently, without the need for assistance. These positive outcomes have implications for greater participation, independence, and control over oneself and the immediate environment. It was difficult to distinguish between the ICF domains of body structure and function, and thus, the results were reported together to include the physical change that occurred, as well as how the changes affected activity engagement and overall health. 

The psychological health outcomes reported also support previous HIFT research findings among non-disabled populations, and include decreased depression and anxiety, heightened feelings of enjoyment, empowerment, and autonomy [23,76]. Unique to these participants include the improvements in self-confidence, the acquisition of coping strategies used outside of HIFT, and participating in HIFT to relieve stress. These psychological health factors have been found to play an important role in sustained engagement, which is particularly important for people with MRD, as research findings indicate that many of the health benefits only occur after long-term exercise participation [1,18]. 

Themes related to psychosocial health outcomes following HIFT participation were broken down into four sub-themes: general social support and affiliation; relationships with HIFT trainers; social media connections; and inclusive HIFT competitions. The HIFT objective to support a sense of community is evident from previous research findings among non-disabled populations [34,72], and the themes were prevalent among these participants, who seem to have formed their own community within the greater HIFT community. Through social media connections where adaptive functional movement strategies could be shared and camaraderie is formed between individuals with similar impairments, through the growth of HIFT competitions for participants with disability, and through the ATA, many of these psychosocial themes appeared to be participant-driven. Other contextual factors specific to HIFT were also found to promote psychosocial connections and health, including integrated class settings and the emergence of HIFT trainers with disability. The community-building aspect of HIFT supports components of behavioral change theories (e.g., self-determination theory; social cognitive theory), and has been found to positively influence exercise adherence for people with disability [4,37]. 

Additional themes were identified related to contextual factors from the HIFT environment that promote inclusion and accessibility for participants. The need for novel equipment designed specifically for people with MRD is minimal but may include devices such as lap mats for seated individuals, a wide base for wheelchairs, or assistive devices for grip support. Most HIFT activities occur in open indoor spaces, which are often almost completely accessible. These features directly address the accessibility barriers often cited by people with disabilities for community-based exercise settings [11]. 

Participants reported a strong desire to share their HIFT experience and health outcomes with the disability and healthcare communities—an important contribution to the literature on exercise for people with disabilities. Participants describe changes to clinical indicators of health, slowing of disease progression, and a desire to include the functionality incorporated in HIFT as part of, or in conjunction with, short-term rehabilitation programs. This unique theme has implications for healthcare approaches typically utilized to treat secondary conditions (e.g., surgical and pharmaceutical), as these findings provide evidence of a community-based exercise program that can be utilized to manage or prevent secondary conditions for people with MRD [15]. HIFT challenges the attitudinal barriers that people with disability often experience, where medical services are more specialized as opposed to integrative [77], as it has the potential to provide a sustainable and beneficial alternative to medical interventions. As one participant explained:

There is a general feeling in our society that those who are disabled are less than, are broken, and must be protected. That we might break if we exercise. Quite the opposite, HIFT and the community that it provides should be considered imperative rather than dangerous. While our exercise might look different than [people without disability], it is what allows us to be successful in life. It makes us safer. It gives us confidence. It slows the progression of our illness or even stops and turns around deterioration.

Collectively, these findings describe the subjective health effects of a community-based exercise program that reduces disparities in access to, and engagement in, exercise for people with MRD. Given the abundance of HIFT facilities across the U.S., vast potential exists for the dissemination and reach of these programs to expand their membership base and include individuals with MRD from their communities. The identified contextual factors could also serve as a model for how non-HIFT programs could be more inclusive to people with disability in their community. The findings also support the mobilization of this knowledge into disability and healthcare communities. 

## 5. Limitations and Future Research

The study design is limited by the retrospective approach taken to examine the health effects of HIFT as opposed to a pre–post design [60]. However, a pre–post assessment of subjective health outcomes may result in response-shift bias due to an inaccurate frame of reference, or lack of knowledge about oneself prior to the program implementation [78,79]. For example, assessing transfer difficulty and satisfaction prior to an intervention may result in low levels of perceived difficulty. At the end of the intervention, the participant may have a new perspective on the ease of transferring due to strength and mobility gains, and indicate low difficulty post intervention, suggesting an ineffective intervention. Thus, we employed a retrospective evaluation to gain a more accurate assessment of perceived health changes due to HIFT engagement [59,60,78,80]. Future research efforts should consider including perspectives from former HIFT participants with disability who, for whatever reason, are no longer participating in HIFT [79]. The reasons for ceasing HIFT participation would provide greater insight into how HIFT affiliates could better support participation for people with MRD. Finally, the self-selection bias from convenience sampling may have resulted in more enthusiastic HIFT participants with more positive experiences than others [80].

Barriers inherent to community-based programs remain despite the inclusive and accessible environment of the program, including commonly cited challenges related to transportation, time, or financial constraints [5,6,7,9,12]. Some, but not all, current HIFT programs are able to offer sliding scales, reduced service fees, or scholarships to support memberships for people with disabilities. Thus, evidence on the effectiveness of this program is necessary to support program participation and facilitate long-term participation. The alternative strategies for remote HIFT would reduce barriers related to physical access, time, and transportation for people with disabilities. Other barriers remain, however, including the need for individualized modifications of the workouts and supervision, especially in the case of heightened fall risk status for individuals with MRD. Additional research is needed to assess which steps, if any, have been taken to counter such barriers for members with disability within the currently operational HIFT communities.

The findings of this study suggest further exploration is warranted on the effects of HIFT for people with disability. Randomized HIFT interventions with sufficient sample sizes that assess objective changes to physical and functional health are needed, as well as examination into other dimensions of health, such as stress, sleep, or depression. Furthermore, it could be informative to examine the types of movements within HIFT that are most beneficial for increasing functionality among the various types of mobility limitations. An examination of the effects of an extended HIFT program on a post-injury population (e.g., SCI) could produce knowledge about its role as a preventive measure against chronic conditions, rather than the view health professionals sometimes hold as a temporary therapeutic treatment. A future study of health professionals’ current understanding and attitudes about HIFT would inform actions aimed at correcting misconceptions and increasing their recommendation of this form of exercise.

## 6. Conclusions

The findings presented here provide initial evidence of the effects of HIFT participation for individuals with MRD, and describe the strategies implemented within HIFT environments that promote inclusiveness and accessibility. The findings also amplify the voices of current HIFT participants who have opted to initiate and sustain engagement in HIFT, and propose advice for the healthcare and disability communities regarding their HIFT experience. Further research is needed to evaluate objective changes to health following HIFT engagement as a means of preventing or managing secondary conditions for individuals with various types of disabilities. 

## Figures and Tables

**Figure 1 sports-11-00116-f001:**
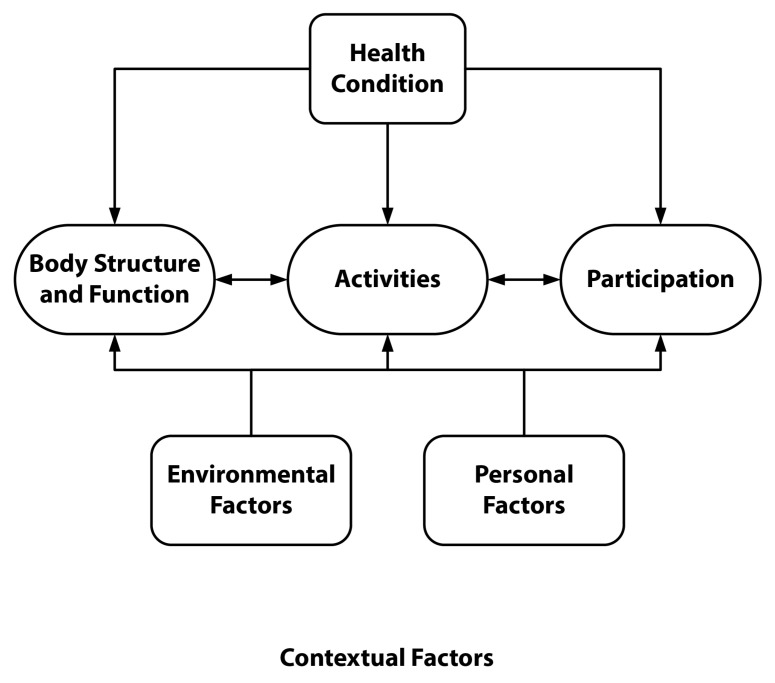
World Health Organization’s International Classification of Functioning, Disability, and Health conceptual diagram [55].

**Figure 2 sports-11-00116-f002:**
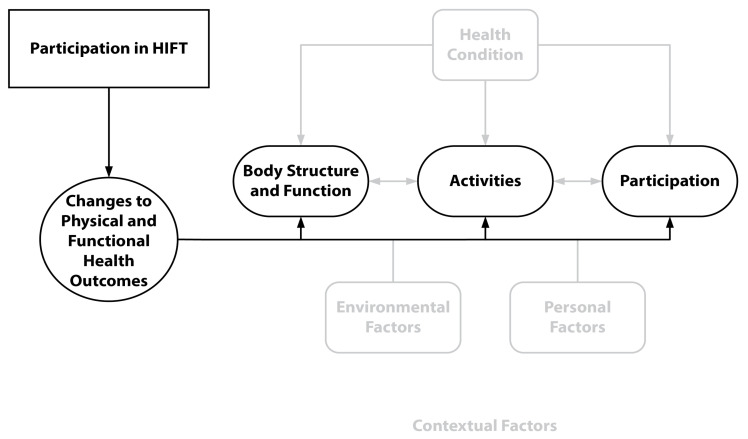
WHO’s ICF Conceptual Diagram with Physical and Functional Health Themes.

**Figure 3 sports-11-00116-f003:**
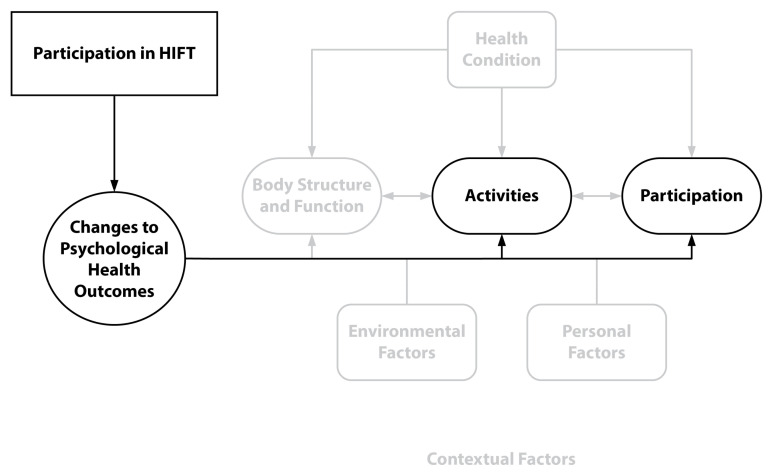
WHO’s ICF Conceptual Diagram with Psychological Health Themes.

**Figure 4 sports-11-00116-f004:**
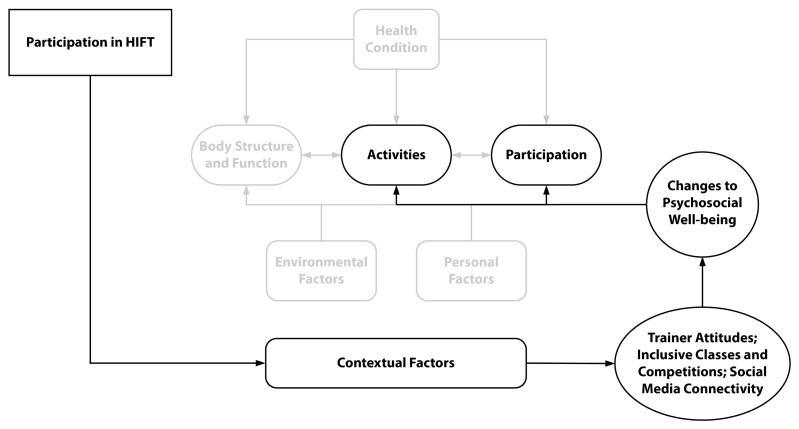
WHO’s ICF Conceptual Diagram with Psychosocial Themes.

**Figure 5 sports-11-00116-f005:**
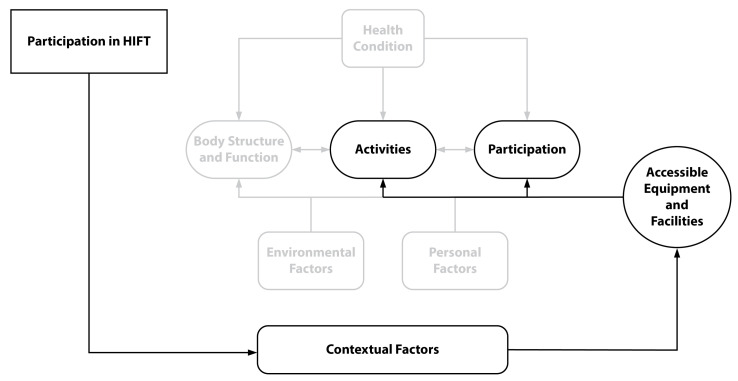
WHO’s ICF Conceptual Diagram with Environmental Themes.

**Table 1 sports-11-00116-t001:** Participant Descriptive Characteristics (N = 38).

Characteristic	Survey Only (*n* = 28)		Interview + Survey (*n* = 10)	
	*n*	%	*n*	%
Gender				
Female	15	54	6	60
Male	13	46	4	40
Age				
*M =* 37.7 years (*SD* = 9.3) Range = 25–58			*M =* 45.3 years (*SD* = 12.6) Range = 26–62	
18–34 years	15	54	2	20
35–64 years	13	46	8	80
Race ^a^				
White race	13	93	9	90
Not reported	1	7	1	10
Ethnicity ^a^				
Hispanic/Latinx	2	14	0	0
Non-Hispanic/Latinx	12	86	9	90
Not reported	0	0	1	10
Employment ^a^				
Full time	13	93	4	40
Other/not specified	1	7	6	60
Primary Disability Category				
Congenital (e.g., limb difference)	5	18	1	10
Health impairment (e.g., rheumatoid arthritis)	0	0	1	10
Injury (e.g., spinal cord injury, amputation)	13	46	4	40
Neurological (e.g., cerebral palsy)	10	36	4	40
Age of disability onset ^b^			*M* = 23.4 years (SD = 22.2) Range birth to 56 yrs.	
Length of time of having primary disability ^b^				
			*M* = 21.9 years (SD = 17.1) Range, 1–56 yrs.	
Length of HIFT participation				
	M = 4.2 yrs.; SD = 2.9 Range, 3 mo. to 12 yrs.		M = 3.3 yrs.; SD = 2.4 yrs. Range, 7 mo. to 9.1 yrs.	

Notes: HIFT—high-intensity functional training. ^a^ Only 14 participants in the Survey-Only participant group (*n* = 28) responded to this category. ^b^ Only Interview + Survey participant group (*n* = 10) responded to this category.

**Table 2 sports-11-00116-t002:** Physical and Functional Health Quotes by Theme.

Subtheme/Quotes (Participant Identifier; Disability Category)
*Physical Health*
“My blood sugar was never high anymore. My A1C’s down to 5.3, which is normal.” (AA-02, injury) *
“Avoided physical therapy by strengthening my whole body through [HIFT]” (AA-53, congenital)
“I have a rare disease called Klippel–Trenaunay syndrome, [HIFT] has allowed me to gain muscle I was not born with and also keep my disease maintained from progressing. Weight loss and even success with my surgeries and treatments. My heart health has improved, and it calms my nerve pain and improves its function.” (AA-68, congenital)
“Every time I go to the doctor I get told that I have the best vitals/blood work they’ve seen all day.” (AA-06, neurological)
“[Increased] strength to lift, carry, or drag items” (AA-55, congenital)
“Since starting [HIFT], my MS symptoms have minimized. My gait and balance have drastically improved. I am no longer falling. My walking speed is much faster. I also experience less fatigue and brain fog” (AA-01, neurological)
“Health-wise it has given me more stamina, more strength. It has corrected a curving spine” (AA-05, injury)
“I have better balance and strength. More than I ever have had. I have had 8 surgeries (joint replacements & fusions) due to my RA [rheumatoid arthritis]. I have osteoporosis and I have COPD [chronic obstructive pulmonary disease], but I feel GREAT. I can do things I never thought I could, like bench presses, wall balls, etc. The coaches work with me to adapt to my physical limitations” (AA-04, health impairment)
*Functional Health*
“But I think what [HIFT] has helped, is if I would unload the dishwasher, it would take me hours to recuperate, or maybe a day to recuperate from doing the dishes. And now, I’m cooking, I’m cleaning up afterwards, I empty the dishwasher.” (AA-02, injury) *
“It has given me more energy and helped me be more independent on my own. Example: I can now put my chair in and out of my car on my own. And I can stand assisted for longer periods of time.” (AA-06, neurological)
“Opening doors. I always used to struggle opening door handles. But I’ve got the ability now. And the technique of being able to actually open the door, door handle and pull the door, shut the door. Obviously, just wheeling around the house is a lot easier than it used to be.” (AA-05, injury) *
“[HIFT] has allowed me to take care of some of my own needs without so much relying on somebody else.” (AA-02, injury) *
“I think [HIFT] is a great path to getting that autonomy. Anytime you can get more function, more strength, and you’re not relying on somebody else, you have more options.” (AA-03, injury) *
“I couldn’t do wheelchair ramps before. And now I go up wheelchair ramps like they’re flat.” (AA-02, injury) *
“I credit [HIFT] for being able to continue working, and for my ongoing general physical and mental wellness.” (AA-09, neurological)
“[HIFT has] been especially helpful in my daily life as a wheelchair user. My strength is important to do transfers properly and my endurance is important to continue to live independently.” (AA-51, injury)
“I am much stronger, have better balance, and better ability to live my life the way I want. I strongly believe [HIFT] has played a major role in my strength and independence, at this point as my disease progresses.” (AA-09, neurological)
“Improves strength, balance, and endurance, which affects my capability to improve and accomplish all ADLs [activities of daily living].” (AA-59, injury)

Note: * = interview response.

**Table 3 sports-11-00116-t003:** Psychological-Health-Themed Quotes.

Quotes (Participant Identifier; Disability Category)
“For me it’s mental, the physical aspect is secondary.” (AA-106, congenital)
“And to increase my self-esteem and confidence. I am learning how to know my own body within by knowing when to push, and when not to.” (AA-58, neurological)
“[HIFT] has helped me believe in myself again unlike anything else I’ve done.” (AA-101, Injury)
“Outside the gym, [HIFT] has given me many coping strategies to handle day to day life.” (AA-09, neurological)
“[HIFT] has given me tremendous confidence again, [HIFT] seems to be a family and not just people at the gym.” (AA-02, injury) *
“I’m 40 years old. I lived 38 of those years lacking confidence, and self-esteem because I’m different. [HIFT], specifically my [gym] gave me an opportunity to shine and be my best self.” (AA-07, congenital) *
“Significant reduction in fatigue and depression.” (AA-01, neurological)
“[HIFT] keeps me happy and sane. My job can be very stressful. And working out at the end of the day is a stress reliever. It refreshes me and energizes me.” (AA-09, neurological)
“Helps with anxiety and depression and PTSD [post-traumatic stress disorder].” (AA-106, congenital)

Note: * = interview response.

**Table 4 sports-11-00116-t004:** Psychosocial Well-being Quotes by Subthemes.

Subtheme/Quotes (Participant Identifier; Disability Category)
*General Social Support/Affiliation*
“The social aspect of the [HIFT] community is the best around. I work out 4–5 days a week at my gym as the only adaptive athlete. Nobody there cares or treats me differently outside of offering help to get gear set up.” (AA-08, injury)
“[HIFT] provided a sense of community and comradery that really helped me at the time. The gym was very inclusive and supportive of me.” (AA-103, Injury)
“And it’s more of a family the way people cheer you on and root you on, and they want to see you succeed. It’s not like going to the gym—it’s not like that. There’s a whole community [in HIFT]” (AA-02, injury) *
“[HIFT] has also made my confidence skyrocket thanks to the ability to do workouts with other people. Even though I might have a shorter run or do step-ups instead of box jumps, the workouts still feel like I’m working with the group and pushing myself.” (AA-50, congenital)
“The social support of the community has been huge, and my ability to be around other adaptive athletes has been important in my mental and physical health.” (AA-09, neurological)
“As a person with a spinal cord injury who lives all day in a wheelchair, I am always viewed differently by the general public. They feel that we are weak, always need help, and someone to feel sorry for. The HIFT community helps show you none of that is true. That is a huge benefit to anybody’s mental health.” (AA-08, injury)
“They’ve never looked at me as just a person in a wheelchair. I’m just one of I am one of the members.” (AA-05, injury) *
*Relationships with HIFT Trainers*
“The owner of our gym messaged back to me was like, ‘Hey, I’ve never worked with a person in a wheelchair, but I’ll give it a try. Come on in.’ In my 4 years of doing [HIFT] I have had zero injuries since I have coaches that make sure I do not overuse certain muscles. They make sure that my workouts stay varied, so nothing is overused.” (AA-08, injury) *
“Unlike a lot of places where, when I roll in immediately the owners are nervous that I’m setting them up for some sort of action under the Americans with Disabilities Act. And in the [HIFT] box, [they] were welcoming us. There weren’t any coaches that you could sense hostility from. They all seem to embrace us being there.” (AA-03, injury)
“[My HIFT] has an incredible coaching staff with a decade of programming and [HIFT] coaching experience. They have navigated my needs as an adaptive athlete and have a lot to do with all of the progress I’ve made since joining their gym in August 2019.” (AA-60 neurological)
“Especially with being a quadriplegic and having more weight on me than I wanted, it’s really not easy to get in and out of your chair. [My trainer] had never worked with a person in a wheelchair before, but he was all for learning. And I’ll never forget, he treated me no different than he does any other new member who comes in—he literally had me do the exact same workout that he has everybody else do. He actually had me getting out of my wheelchair, which I hadn’t voluntarily done in probably 10 years.” (AA-08, injury) *
*Social Media*
“Through Instagram I found other adaptive athletes doing [HIFT] and that was the most eye-opening experience. I’ve made friends with other adaptive athletes by sharing our stories and how we adapt to different movements.” (AA-07, congenital) *
“The friends I have made from [HIFT Competitions] and through social media that are all in the same situation as me is great.” (AA-08, injury)
“So, I first got started [on Instagram]. And I found all these really great athletes like [names omitted], they both have missing limbs. And I was like, whoa, wait, this kid is climbing a rope. I found all these other really amazing athletes that were just like me, and I’m like, ‘Alright, so I can do this!’” (AA-07, congenital) *
*Inclusive Competitions*
“Being able to compete even with adaptive movements has given me the chance to train with people and not be excluded.” (AA-05, injury)
“Through the recent announcement of including adaptive athletes—I have the chance to compete with others just like me. I’ve never been able to do that. Ever.” (AA-07, congenital) *
*Integrated Classes*
“[HIFT] is more than just about working out and getting fit. It is about a community of people who support each other. It’s a group of people who all look to me as an equal and not a disabled person. That helps everyone’s mental health but especially people with a spinal cord injury since we are viewed differently.” (AA-08, injury)
“No, we just have a really special [gym], because it has an adapted focus, and I call it the land of misfit toys because we kind of joke that we assume everyone is adaptive until proven differently.” (AA-01, neurological) *
“Even though I am an adaptive athlete, people treat me and push me the same as any other athlete.” (AA-06, neurological) *

Note: * = interview response.

**Table 5 sports-11-00116-t005:** Environmental Factors-Themed Quotes.

Quotes (Participant Identifier; Disability Category)
“I have a custom-made jump rope. It’s got an attachment that goes around my left arm that can be tightened. And so that I jump right when I jump rope, it looks very much like an able-bodied person jump roping.” (AA-07, congenital) *
“I’ve made my own wheelie bar, that is a very heavy-duty wheelie bar. So that whenever I’m lifting up anything overhead, I can’t flip or anything else to hurt myself.” (AA-08, injury)
“[The box I’m at now], he purposefully designed it so it could be wheelchair accessible.” (AA-01, neurological) *
“We’ve gone through a couple variations of attachment to the rig. What I currently use is a construction strap, looping around the rig and I can jump into it.” (AA-07, congenital) *
“And so, whenever the gym got a SkiErg, I of course cannot fit in my wheelchair. So, I took two old belts that I had that were Canvas belts looped them around the handles, then use those as kind of extensions.” (AA-08, injury) *

Note: * = interview response.

**Table 6 sports-11-00116-t006:** Advice-to-Others-Themed Quotes.

Quotes (Participant Identifier; Disability Category)
“I think [HIFT] should have been introduced to people of all disabilities a long time ago. It should be a coordinated effort between healthcare and [HIFT] training.” (AA-04, health impairment)
“I have been told over and over again by doctors that [HIFT] is dangerous, that I shouldn’t do it, that I won’t be able to do many of the movements. Quite to the contrary, over and over I prove them wrong. While my body requires constant maintenance and my capabilities are changing as my condition progresses, [HIFT] has given me the strength and the tools to maintain a high level of health and fitness. Doctors should spend more time understanding what [HIFT] actually is, and the positive benefits (physical and mental) of doing it and being a part of the community.” (AA-09, neurological)
“As soon as I started doing [HIFT], I thought it should be included in the recovery of any spinal cord injury program.” (AA-51, injury)
“I would like for healthcare providers to know—when done correctly and adapted with a knowledgeable coach—the physical and emotional benefits of [HIFT] far outweigh any medication.” (AA-10, neurological)
“[HIFT] is/was the single most important contributory factor in my recovery from amputation.” (AA-72, injury)
“Doctors don’t understand how [HIFT] is important with my neurological problem and I’m really frustrated that healthcare system does not understand how good it is.” (AA-74, neurological)
“Each person can start out with traditional therapy mixed with some [HIFT] and gradually work towards exclusively [HIFT] that can be maintained once they leave rehab and are at home.” (AA-59, injury)
“I don’t really think that there’s any disability out there that wouldn’t benefit in some way from [HIFT].” (AA-06, neurological) *
“[HIFT] has kept me functional and well far better than any experience I’ve had with doctors or physical therapy. My neurologist is consistently impressed by my strength, independence, and balance.” (AA-09, neurological)

Note: * = interview response.

## Data Availability

Data sharing not applicable.

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
