# Peer review of "High-Intensity Functional Training: Perceived Functional and Psychosocial Health-Related Outcomes from Current Participants with Mobility-Related Disabilities"

_sports, 2023, doi:10.3390/sports11060116_

Round 1

Reviewer 1 Report

This paper discusses the benefits of exercise, par5cularly High-Intensity Functonal Training (HIFT), for individuals with mobility-related disabili5es (MRD). It highlights the lack of research on community-based exercise programs for individuals with MRD and the barriers they face in partcipatng in such programs. The paper emphasizes the inclusive nature of HIFT, which allows partcipants to modify workouts based on their needs and capabili5es. It also highlights the importance of the social environment in promotng partcipaton and adherence to exercise programs.
The study described in the paper aims to evaluate the perceived health outcomes and moves for engagement among current HIFT partcipants with MRD. The findings of the study indicate improvements inphysical, functonal, psychological, and social health among partcipants. Partcipants reported improvements in body compositon, strength, mobility, and engagement in everyday actvites. They also experienced psychological benefits such as decreased depression and anxiety, increased enjoyment, empowerment, and
autonomy. The social environment of HIFT, including social support, rela5onships with trainers, social media connectons, and inclusive competitons, was found to promote psychosocial health.
In my opinion, there is a lack of critcal discussion on potental risks and limitatons: The artcle primarily focuses on the perceived benefits and positve outcomes of HIFT for individuals with MRD. However, it does
not cri5cally discuss potental risks, challenges, or limita5ons associated with HIFT. It is important to address factors such as the level of physical exerton, risk of injury, and the need for individualized modificatons and supervision in HIFT programs for individuals with MRD.
Despite not being the focus of the artcle, I think it should be addressed the socioeconomic factors in the discussion. While the artcle briefly mentons barriers to partcipaton, such as transportaton or financial constraints, it does not extensively discuss the influence of socioeconomic factors on accessibility and engagement in community-based exercise programs for individuals with MRD. Addressing these barriers is crucial to ensure equitable access and inclusion for all individuals, regardless of their socioeconomic status. In conclusion, this paper provides inital evidence of the positve effects of HIFT partcipaton for individuals with MRD. It emphasizes the inclusive and adaptable nature of HIFT and its poten5al to improve various aspects of health and well-being. The findings highlight the importance of community-based exercise programs and call for further research and dissemina5on of knowledge to promote accessibility and inclusion in exercise for individuals with MRD. Therefore, while non-disabled populaton studies can provide some insights, it is crucial to prioritze research specifically conducted on individuals with MRD to understand the impact of HIFT on their health and well-being.In page 14, line 55 "As one participant explained,

There is a general feeling in our society that those who are disabled are less than, are broken, and must be protected."

suggestion - should be written this way - "As one participant explained, there is a general feeling in our society that those who are disabled are less than, are broken, and must be protected."

Author Response

We greatly appreciate this reviewer's comments and considered each point thoughtfully. Overall, the interview script was designed to explore participant experiences and changes to health outcomes related to HIFT participation. While HIFT does address some of the common barriers inherent to in-person exercise for people with MRD, it certainly doesn't address all. Thus, we have added a paragraph to the limitations section addressing this fact with suggestions for future research to explore how participants and/or adaptive HIFT communities have addressed such barriers (successfully or unsuccessfully), particularly in regard to financial constraints. It is out of the scope of this study and manuscript to discuss meaningfully potential risks associated with HIFT engagement, as risks are a part of any exercise program, regardless of mode and context, for people with and without disability. We do however, suggest that future research recruit and explore the experiences of people with MRD who are no longer participating in HIFT to further learn from their experiences and options for ceasing participation.

Reviewer 2 Report

This is a very interesting paper aiming to explore the experiences of adults with mobility-related disabilities who currently participate in high-intensity functional training, an inclusive and accessible community-based exercise program.

The article is well written and reader-friendly, adding merit to an undiscoverable research area. However, such a training modality is considered an attractive option among exercisers across the world.

Thus, this article provides novel insights into the implementation of CrossFit-based training programs for those experienced mobility-related disabilities in the real world. In summary, I have no concerns regarding the quality and novelty of this work.

The only minor comment is about an additional reference highlighting the current top trends in the fitness industry (1), including those highly involved in CrossFit-like programs.

1. Kercher et al. ACSM's Health & Fitness Journal 2023; 27(1): 19-30. DOI: 10.1249/FIT.0000000000000836

Author Response

Thank you for taking the time to review this manuscript.  We appreciate the updated citation, and have replaced Thompson et al., 2019 with the Kercher et al., 2023 article you suggested.